# Enhancing Wheat Gluten Content and Processing Quality: An Analysis of Drip Irrigation Nitrogen Frequency

**DOI:** 10.3390/plants12233974

**Published:** 2023-11-26

**Authors:** Tianjia Hao, Rong Chen, Jing Jia, Changxing Zhao, Yihang Du, Wenlu Li, Ludi Zhao, Hongxiao Duan

**Affiliations:** 1Shandong Provincial Key Laboratory of Dryland Farming Technology, College of Agronomy, Qingdao Agricultural University, Qingdao 266109, China; haotianjia20230313@163.com (T.H.); cr03010226@163.com (R.C.); jiajing_wy@163.com (J.J.); yihangdu0520@163.com (Y.D.); 17863951027@163.com (W.L.); z18053902431@163.com (L.Z.); d19894332609@163.com (H.D.); 2Key Laboratory of Agro-Environment in Downstream of Yangtze Plain, Ministry of Agriculture Institute of Agricultural Resources and Environment, Jiangsu Academy of Agricultural Sciences, Nanjing 210014, China; 3College of Agronomy, Nanjing Agricultural University, Nanjing 210095, China

**Keywords:** drip irrigation nitrogen frequency, wheat, dough rheological properties, CIE L*a*b*

## Abstract

Drip irrigation is a water-saving and fertilizer-saving application technology used in recent years, with which the frequency of drip irrigation nitrogen application has not yet been determined. In order to investigate the effects of different drip irrigation nitrogen application frequencies on the processing quality of medium-gluten wheat (Jimai22) and strong-gluten wheat (Jimai20 and Shiluan02-1), a two-year field experiment was carried out. Two frequencies of water and N application were set under the same conditions of total N application (210 kg·ha^−1^) and total irrigation (120 mm): DIF4, consisting of four equal applications of water and N (each of 30 kg·ha^−1^ of N application and 30 mm of irrigation) and DIF2, consisting of two equal applications of water and N (each of 60 kg·ha^−1^ of N application and 60 mm of irrigation). The results showed that IF4 significantly increased protein content by 2–8.6%, wet gluten content by 4.5–22.1%, and hardness value (*p* > 0.05), and PC2 was considered as a protein factor; the sedimentation value was highly significantly correlated with most of the parameters of the flour stretch (*p* < 0.01). DIF4 improved the stretching quality, and the flour quality of Jima22 was decreased, the flour quality of strong-gluten wheats Jimai20 and Shiluan02-1 was improved, and PC1 was considered to be the dough factor. In conclusion, although the frequency of nitrogen application by drip irrigation increased the protein factor and improved the tensile quality, the flour quality was not necessarily enhanced.

## 1. Introduction

Nitrogen fertilizer plays a crucial role in the continuous increase in wheat yield in China [1,2]. It has been difficult to improve yield using traditional fertilization methods [3,4], and it is increasingly important to improve quality and efficiency. The sustainable development of the wheat industry requires the synergistic improvement of wheat yield and quality, so the improvement of grain quality while emphasizing the improvement of yield has become an urgent problem in China [5]. Nowadays, people are gradually changing from “eating enough” to “eating well”, and there is an urgent need for field cultivation to realize the synergistic improvement of grain yield and quality [6,7].

China’s traditional mode of irrigation is border irrigation (heavy flooding of fields with ridges), and the mode of fertilizer application is to spread fertilizers (sprinkle fertilizer on the surface of the field), resulting in serious losses of water and nitrogen and harming the environment [8,9]. China feeds 21% of the world’s population with 9% of its arable land and 6% of its fresh water, and limited resources and environmental sustainability are severely constraining the supply of agricultural products [9]. Water–fertilizer integration technology originated in Israel in the 1960s, and drip irrigation equipment was introduced in China in the 1970s, and is now in the stage of vigorous promotion and application in China [10]. Drip irrigation realizes the coupled application of water and nitrogen into the soil at the root system [11], which not only improves the efficiency of water and nitrogen utilization and reduces the irrational use of resources [12], but also protects the environment and reduces N wetting [13] and C emissions [14].

Nitrogen is an important factor affecting the protein and wet gluten content of wheat [15], and the quality of wheat dough and pasta is largely influenced by protein content and gluten quality [16,17]. Changes in grain color may affect the color of the flour and the final product and may be influenced by changes in other grain traits [18]. Hardness is related to protein content [19,20]. The rheological properties of wheat have improved significantly since 1986, while the flour properties have remained relatively stable [21]. Xue et al. downscaled the relevant quality traits of wheat and concluded that flour quality is mainly influenced by the rheological properties of dough [22]. Gliadins and glutens comprise the GSPs (Grain Storage proteins), which are the main body of gluten [23,24] and together account for 60–80% of the total seed proteins, determining the elasticity (tensile area) and ductility of the dough, and determining the processing qualities of the various end-products [25]. Nitrogen treatment alters the expression levels of genes involved in storage protein production [26]. Wheat gluten proteins have a high molecular weight and form a network skeleton that provides gluten strength (resistance to deformation) as well as dough elasticity [27,28], which is decisive for the rheological properties of dough [29]. Sedimentation values show a highly significant positive correlation with quality traits [30] and can be used as a comprehensive indicator of quality [21].

The net productivity of N from jointing to flowering was significantly correlated with yield [31]. Rational regulation of leaf senescence can improve yield and quality by influencing photosynthesis and nutrient remobilization [32], and less research has been conducted on water N application during the irrigation period. Whether split application of nitrogen fertilizer through drip irrigation facilities can further improve wheat quality is yet to be investigated. It is particularly important to establish crop management measures for drip irrigation fertilization. In this study, different varieties of medium-gluten and strong-gluten wheat were selected by setting up high and low frequencies of drip irrigation nitrogen application, comparing the differences in processing quality and exploring the following: (1) the effects of drip irrigation nitrogen application frequency on the dough quality traits of medium-gluten and strong-gluten wheats and (2) the relationship between the dough quality traits.

## 2. Results

### 2.1. CIE, Hardness Value, Protein Content, Wet/Dry Gluten Content, Sedimentation Value

Year had a significant effect (*p* < 0.05) on CIE a*, and a highly significant effect (*p* < 0.01) on CIE L*, CIE b*, protein content, and gluten content (Table 1), but not on hardness value and settling value. The hardness values in this study were not affected by the variety factor, probably because the hardness of these three varieties was similar. All these traits were highly significantly affected (*p* < 0.01) when only water nitrogen frequency was considered, and it is necessary to regulate seed appearance color CIE L*a*b* (Commission Internationale d’Eclairage LAB color space), hardness attributes, protein content, gluten content, and sedimentation value by improving water nitrogen frequency. There were highly significant (*p* < 0.01) changes in protein content and gluten content when year, variety, and frequency interacted.

The protein content of the three varieties with high frequency of nitrogen application was significantly higher than that with low frequency by 2–8.6% (Figure 1), and applying nitrogen fertilizer at all critical periods instead of separately at the nodulation and flowering stages resulted in higher protein content. Shiluan02-1 had the highest protein content (14.3–17.2%), followed by Jimai20 (12.8–15.3%) and Jimai22 (12.8–14.1%). The protein content of the second year was lower than the first year, which might have been antagonized by the yield [33,34], which was 43% higher than the first year as shown in our previous study [35]. High frequency of nitrogen application increased the hardness of the grains, but it was possible that the large differences between the grains resulted in no significant differences between treatments (Figure 2). High frequency of nitrogen application reduced the brightness of wheat kernels, increased redness, and reduced yellowness; CIE L* was lower in DIF4 than in DIF2; CIE a* was higher than in DIF2; and CIE b* was lower than in DIF2 (Figure 3).

Wet gluten content was significantly increased by 4.5–22.1% under high-frequency drip irrigation with nitrogen application, and the difference in dry gluten content of Jimai20 in the second year was not significant, with the three varieties increasing by 7.1–37% (Figure 4). The wet gluten content of Jimai20 and Shiluan02-1 varied little between the two years, and the wet gluten content of Jimai22 in the second year was higher than that of the first year, although the gluten index of Jimai22 in the second year was lower than that of the first year by 17.6–28.3%. Although the wet gluten content of Jimai22 was slightly higher than that of Jimai20 and Shiluan02-1, the gluten index was lower than that of both. High-frequency nitrogen application decreased the sedimentation value of Jimai22 and increased the sedimentation value of Jimai20 and Shiluan02-1 (Figure 5). There was no significant difference in the sedimentation values of Jimai22, except for Shiluan02-1 in the second year, where the sedimentation values of DIF4 were significantly higher than those of DIF2 by 3.1–12.8%.

### 2.2. Flour Quality

Year had a significant effect (*p* < 0.05) on stabilization time (Table 2). Variety had a significant effect (*p* < 0.05) on water absorption, and both had highly significant effects (*p* < 0.01) on the other flour quality traits, respectively. Different frequencies of nitrogen application had a highly significant effect (*p* < 0.01) on stabilization time of dough and a significant effect (*p* < 0.05) on FQC; frequency of nitrogen application can affect flour quality. When the three factors interacted with each other, the stabilization time, moisture content, and tolerance index were affected, and the final flour quality was also affected.

Medium-gluten wheat Jimai22 had higher flour quality under two drip fertilization applications, while the opposite was true for strong-gluten wheats Jimai20 and Shiluan02-1 (Table 2). In both years, the FQC (flour quality index) of DIF4 was significantly 23–80% higher than that of DIF2 for the strong-gluten varieties Jimai20 and Shiluan02-1, with the exception of Shiluan02-1 in the second year, which had higher stability time and development time, and lower values of tolerance index as well as a decrease in consistency after 10 min. The FQC of DIF4 of Jimai22 was lower than that of DIF2, but there was no significant difference. In both years, the water absorption, development time, and stability time of Jimai22 were higher for DIF2 than for DIF4, and the degree of weakening after 10 min was higher for DIF4 than for DIF2, while the opposite was true for the strong-gluten wheats Jimai20 and Shiluan02-1. This may be due to the fact that the gluten index of DIF4 was lower than that of DIF2 in Jimai22, which reached a significant level in the second year (Table 3). The gluten indexes of strong wheats Jimai20 and Shiluan02-1 were close to 100%, and there was no difference between the different drip irrigation N application frequencies.

### 2.3. Tensile Quality

Frequency of water nitrogen application had significant (*p* < 0.05) effect on tensile area (Table 4); year, variety and frequency had highly significant (*p* < 0.01) effects on other tensile traits, respectively; and when all the three were inter-cropped, extensibility was highly significantly affected (*p* < 0.01).

High-frequency drip irrigation application of nitrogen increased the tensile area and extensibility of wheat dough, reduced the maximum resistance to extension as well as the resistance value at 5 cm, and improved the tensile quality (Table 4). The extensibility of DIF4 was significantly higher than that of DIF2 by 9.3–23.7%, except for the second year’s Shiluan02-1. The maximum drag of DIF4 of the two years’ Shiluan02-1 was significantly lower than that of DIF2 by 9.4% and 9.9%, respectively, and there was no significant difference in the maximum drag of Jimai22.

The tensile area and maximum drag of Shiluan02-1 were higher than that of Jimai20, and Jimai22 was the weakest. Except for the extensibility of Jimai22 in the first year, the linear fits of extensibility and maximum drag of all three varieties were significant or highly significant above the 0.05 level, with extensibility showing a parabola with an upward opening and maximum drag showing a parabola with a downward opening (Figure 6). The trend of wheat flour tensile quality with time was similar to a parabola.

### 2.4. Principal Component Analysis (PCA)

PCA was used to visualize the variation in rheological properties of wheat by simplifying a large number of traits into PCs, with PC1 and PC2 describing 49.9% and 19.3% of the overall variance, respectively (Figure 7). Powdery stretching traits were strongly positively correlated with PC1, protein content and extensibility were strongly positively correlated with PC2, and CIE L*a*b*of the kernel was strongly negatively correlated with PC2. Compared to DIF4, DIF2 scored lower in PC2 due to lower protein content, extensibility, hardness value, and wet/dry gluten content in DIF2 than in DIF4. In the second year, Jimai20 and Shiluan02-1 scored higher in PC1, and the two-year-old Jimai22 scored negative in PC1. The flour stretch, sedimentation value, and protein content of Shiluan02-1 were higher of the two years.

### 2.5. Correlation Analysis

Positive and significant correlation (*p* ≤ 0.01) was found between CIE a* values and wet/dry gluten content of seeds, while negative and significant correlation (*p* ≤ 0.01) was found between CIE L* and b* values and PC (protein content), HV (hardness value), and E (extensibility), respectively (Figure 8). There was a highly significant positive correlation (*p* ≤ 0.05) between S (Sedimentation) and PC, and a significant correlation (*p* ≤ 0.01) between S and powder stretching parameters, except for WA (water absorption), MC (moisture content), and E, i.e., there was a significant positive correlation (*p* ≤ 0.01) between S and the powder stretching properties. Significant negative correlations (*p* ≤ 0.01) were found between DT (development time) and ST (stability time) and the decreasing values of TA (tensile area) and consistency after 10 min (lowering value in the 20th minute, weakening degree, lowering value in the 20th minute) as well as SR (stretch ratio), respectively, and a significant positive correlation (*p* ≤ 0.01) was found between DT and ST and FQC. Significant positive correlations were found between DT and ST at *p* ≤ 0.01 and stretch parameters except E. Very significant positive correlations were found between stretch parameters except E (*p* ≤ 0.01).

## 3. Discussion

In the actual sale of grain by Chinese farmers, weight and moisture are determined empirically by the buyer, and the actual purchase price is thus determined. Kernel color may be a more intuitive way to determine quality and thus price. Seed color L* and b* were highly correlated and both were negatively correlated with protein content [36,37], and grain color may be closely related to other grain traits [18]. This is consistent with this study where PC (*p* ≤ 0.01) and HV (*p* ≤ 0.05) showed significant negative correlation with L*, a* and b*, indicating that the lower the hardness and protein content, the brighter and yellower the grains were, increasing the frequency of nitrogen chasing with drip irrigation reduced the brightness and yellowness of the grains, and higher frequency of nitrogen application increased the redness of the grains. Grain a* and b* were inversely correlated with quality [36]; a* showed significant negative correlation with ST, TA, and E, but b* was not observed to show significant correlation with flour stretch.

Nitrogen application at the stage of booting to flowering increases the nitrogen content of the nutrient organs and thus ensures more transfer to the seed [38], where 1/3 of the nitrogen in the seed comes from retransfer from the nutrient organs and 2/3 comes from the newly absorbed nitrogen from the soil during the filling stage [39]. The nitrogen fertilizer application at the later stages of growth of the different plants or split nitrogen application is beneficial to increase the protein content [40]. In this study, DIF4 plants were supplied a greater amount of nitrogen at later growth stages and increasing the frequency of nitrogen application increased the protein content and wet gluten content versus dry gluten content. Nitrogen increases the protein content, changes its composition, and affects the rheological properties of dough; higher protein content leads to more stretchable dough [16] and flour from wheat with higher protein content has higher extensibility and lower maximum drag leading to a reduction in the maximum stretch ratio [41]. In this study, by increasing the frequency of nitrogen application through drip irrigation, the TA and E of the dough were increased and the stretch resistance was decreased, which led to a lower stretch ratio and an increase in the tensile quality. Wheat gluten strength varies in different genetic origins, which makes it difficult to determine the effect of protein content [17]. Higher gluten index resulted in higher water absorption, development time, and stabilization time [42]. The gluten index of medium-gluten wheat in this study was opposite to the gluten content, which may be due to the different protein fractions. A higher gluten index resulted in higher dough development time and stabilization time, decreased tolerance index, and increased flour’s flour quality. Numerous studies have demonstrated that higher protein content implies higher gluten proteins, which prolongs the dough development time [43]. In this study, the gluten index of strong-gluten varieties was nearly 100%, while the wet gluten content increased with the frequency of nitrogen application, which resulted in higher WA, DT, ST, and FQC for DIF4 than DF2, and increased flour quality. Numerous studies have shown that the sedimentation value is significantly correlated with other quality traits [30,44], and the sedimentation value can be used as a comprehensive index for evaluating wheat flour milling quality [21,44]. In this study, the sedimentation values were highly significantly correlated with other flour stretching parameters except E, WA, and WC; the flour quality of medium-gluten wheat may be affected by the gluten index; and the sedimentation values can be used as a comprehensive index for evaluating the flour stretching of strong-gluten wheat. Li et al. reported that after downscaling the processing quality of strong-gluten wheat, PC1 was representing the dough factor and PC2 was representing the protein factor [45], and the results of the present study are similar to them.

In fact, this study does not show a synergistic improvement effect of flour quality and tensile properties, although they are significantly correlated. In terms of the water and nitrogen frequency factor alone, high nitrogen application frequency had better tensile quality, and all stretching parameters were significantly affected by frequency, but not all of the flour quality parameters were affected and showed a more stable performance. Yang et al. previously reported that Chinese wheat varieties had improved dough rheological properties with more stable flour quality [21], so could this be attributed to improved stretch quality? In other studies, a consistent increase in tensile quality has been observed when factors are changed, but the corresponding change in flour quality has not been consistent [21,46]. It is difficult to synergize yield and quality at present, although there are some reports on the effects of fertilization and gene regulation on yield and protein content [47], but the improvement of processing quality in fact does not have much relationship with protein content. After all, glutenin determines the processing quality, and the content of gliadins, as a kind of harmful protein, should be reduced. Therefore, reducing the total protein content and increasing the content of glutenin fractions should be an important way to synergize the improvement of yield and quality.

## 4. Materials and Methods

### 4.1. Experimental Design

A 2-year field study was conducted in the Jiaozhou Modern Agricultural Science and Technology Demonstration Park, Qingdao, China (35.53°/N, 119.58°/E) during the winter wheat growing seasons of 2020–2022 on the soil type mortar-black soil, with rainfall of 226.7 mm and 145.1 mm in the two years, respectively. Deep tilling was followed by rotary tillage after straw was returned to the field. The seeds and compound fertilizer (N, P_2_O_5_, and K_2_O were all applied at 90 kg·ha^−1^) were sown by seed and fertilizer co-sowing machines each on 11 October 2020 and 28 October 2021, and wheat was harvested on 17 June 2021 and 18 June 2022. The sowing date of the second year was delayed by successive rainy days. The basic wheat seedlings were about 2.2 million plants-ha^−1^ in rows spaced 20 cm apart. Three rows of wheat were laid with one drip irrigation capillary tube (spaced 60 cm apart) and irrigated using drip heads. A more detailed overview of the pilot site can be found in a previously published article [35].

In this study, the medium-gluten wheat variety Jimai22, and the strong-gluten wheats Jimai20 and Shiluan02-1 were used, with a total nitrogen application rate of 210 kg·ha^−1^ (urea dissolved in water applied) and a total irrigation volume of 120 mm (measured by water meter). Two drip irrigation N application frequencies were set up: high-frequency drip irrigation N application (DIF4) and low-frequency drip irrigation N application (DIF2). In DIF4, water (40 mm) and nitrogen fertilizer (40 kg·ha^−1^) were applied equally at the jointing stage, booting stage, flowering stage, and filling stage, respectively, and in DIF2, water (60 mm) and nitrogen fertilizer (60 kg·ha^−1^) were applied equally at the jointing stage and flowering stage, respectively (Table 5).

### 4.2. Measurements

#### 4.2.1. Preparation of Flour and Whole Wheat Flour

In each plot, 14 m^2^ of wheat was harvested, replicated three times and after maturation, flour was ground using a Laboratory Mill (the manufacturer is CHOPIN, the model number is MOULINCD1, the country is France). Some of the seeds were taken and whole wheat flour was ground using Grain Grinder (Retsch GmbH, TWISTER, German).

#### 4.2.2. Determination of Protein Content

In total, 0.5000 g of whole wheat flour was taken and cooked at 400 °C for one and a half hours using a decoction oven, after which it was cooled for half an hour and analyzed using Kjeldahl Automatic Nitrogen Determination (Gerhardf, VAP50S-K20, German) to obtain the nitrogen content multiplied by a factor of 5.7 to give the crude protein content.

#### 4.2.3. Determination of Gluten Content and Gluten Index

In total, 10.00 g of flour was taken and washed with 2% NaCl solution using the Gluten Washing Device (YUCEBAS-N2, Y07, Türkiye) to obtain wet gluten, weighed and recorded as M_1_, centrifuged using Gluten Index Device (YUCEBAS-N2, Y08, Türkiye), and the weight of the gluten on the tumbleweed was recorded as M_2_. After that, the gluten under the sieve and on the sun screen were dried together in the Gluten Dryer (YUCEBAS, Y09, Türkiye) and recorded as M_3_. This was calculated using the following formula:(1)$$\mathrm{WGC}=\frac{M_{1}}{10}\times 100\%$$
(2)$$\mathrm{GI}=\frac{M_{2}}{M_{1}}\times 100\%$$
(3)$$\mathrm{DGC}=\frac{M_{3}}{10}\times 100\%$$
where WGC (%) is wet gluten content, GI (%) is gluten index and DGC (%) is dry gluten content.

#### 4.2.4. Determination of Sedimentation Value

The sedimentation value of wheat flour was determined using the AACC-56-61-02. In total, 4.00 g of flour (14%) and 50 mL of distilled water were added to a 100 mL stoppered measuring cylinder, which was shaken well and allowed to stand for 5 min. Next, 2–3 drops of methyl blue indicator were added and then 25 mL of lactic acid/isopropanol solution (1:1) was added, after which it was mixed well and allowed to stand for 5 min and then volume of sediment was read as the sedimentation value.

#### 4.2.5. Determination of Hardness Value

The hardness values of single kernels (30 kernels were determined) were determined using the Grain Hardness Tester (Hangzhou Dacheng Photoelectric Instrument Co., GWJ-2, Hangzhou, China) in kg or N; this was repeated 3 times.

Determination of the hardness value of a single grain (30 grains measured) using the Grain Hardness Tester (GWJ-2) was repeated 3 times. The grains were placed on the load table, the top bar was pressurized upwards, and the value displayed when the grains were broken was the hardness value in kg or N.

#### 4.2.6. Powder and Tensile Parameters Testing

Flour stretching quality was determined using a flour stretching machine (YUCEBAS, Y01, Türkiye). In total, 300 g of flour (14%) and 6 g of NaCl were prepared for each sample and water was added (30 °C) within 20 s. The highest peak of consistency during the test was in the range of 500 ± 20 BU indicating accurate data. The powder quality parameters used were water absorption (WA), moisture content (WC), development time (DT), stability time (ST), tolerance index (TI), lowering value in the 10th minute (LV-10th), weakening degree (WD), lowering value in the 20th minute (LV-20th), and FQC. A tensile test was performed at 45 min, 90 min, and 135 min of awakening to determine the tensile parameters tensile area (TA), resistance value at 5 cm (RVF), extensibility (E), maximum drag (MD), stretch ratio (SR), and maximum stretch ratio (MSR).

#### 4.2.7. Determination of CIE L*a*b*

The same volume (about 600–1000 kernels) of wheat kernels was placed in SeedCount trays, and the CIE L*a*b* (Commission Internationale d’Eclairage LAB color space) of the kernels was determined in three replicates by scanning the wheat kernels into an image through reflection using the Seed Image Analysis System (SeedCount, SC6000TR, Australia). CIE L*a*b*: L* was used for brightness with range 0–100 (dark (black)–light (white)), a* for red–green, negative–positive (green–red), b* for yellow–blue, negative–positive (blue–yellow).

### 4.3. Data Analysis

The data were organized in Excel. An analysis of variance (ANOVA) was performed using IBM SPSS Statistics 26 to test the effect of the frequency of water and nitrogen applied to different drip irrigation follow-ups on protein content, gluten content, gluten index, CIE L*a*b*, sedimentation value, flour quality parameters and tensile quality parameters, and interactions between years, varieties, and treatments were examined using reciprocal effects analysis. Differences were compared at *p* < 0.05 using the Waller–Duncan test (Tukey’s HSD test) and plotted using Origin2021pro. Correlations between dough quality traits were analyzed using Pearson’s correlation coefficient.

## 5. Conclusions

Increasing the frequency of nitrogen application by drip irrigation significantly increased the protein content by 2–8.6%, and significantly increased the wet gluten content by 4.5–22.1%. The gluten index of medium-gluten wheat Jimai22 was decreased by 6.5–18.8%. CIE L*a*b* was significantly correlated with protein content and wet/dry gluten content (*p* < 0.01), and increasing the frequency of nitrogen application by drip irrigation decreased brightness and yellowness and increased redness. There was a highly significant correlation between sedimentation value and flour stretching parameters (except water absorption and water content), which can be used as a comprehensive index for evaluating the stretching parameters of gluten wheat flour. Increasing the frequency of nitrogen application by drip irrigation increased the flour tensile quality of strong-gluten wheat, and the tensile quality of medium-gluten wheat improved while the flour quality was decreased, which may have been the effect of the gluten index. After dimensionality reduction, PC1 for processing quality represented the dough factor and PC2 represented the protein factor. In conclusion, the protein and gluten contents can be increased by increasing the frequency of drip irrigation nitrogen application appropriately. In order to obtain higher processing quality, the strong-gluten wheats Jimai20 and Shiluan02-1 are suitable for four drip irrigation applications of nitrogen, and the medium-gluten wheat Jimai22 will obtain higher tensile quality and lower flour quality at four drip irrigation applications of nitrogen.

## Figures and Tables

**Figure 1 plants-12-03974-f001:**
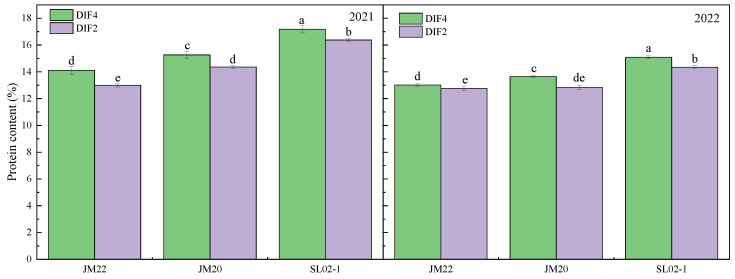
Effect of nitrogen application by high-frequency versus low-frequency drip irrigation on protein content of medium-gluten and strong-gluten wheat. JM22, JM20, and SL02-1 refer to the medium-gluten wheat variety Jimai22, and the strong-gluten wheats Jimai20 and shiluan02-1, respectively. Different lower-case letters indicate significant differences at *p* < 0.05.

**Figure 2 plants-12-03974-f002:**
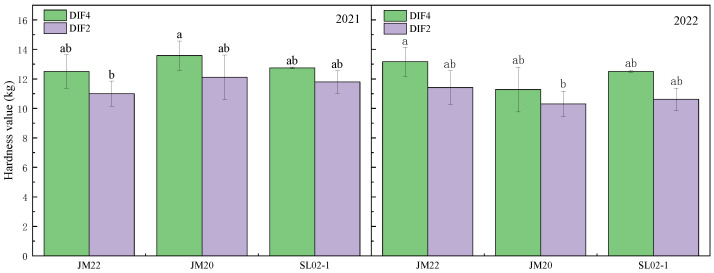
Effect of different frequencies of nitrogen application by drip irrigation on hardness value of medium-gluten and strong-gluten wheat. JM22, JM20, and SL02-1 refer to the medium-gluten wheat variety Jimai22, and the strong-gluten wheats Jimai20 and shiluan02-1, respectively. Different lower-case letters indicate significant differences at *p* < 0.05.

**Figure 3 plants-12-03974-f003:**
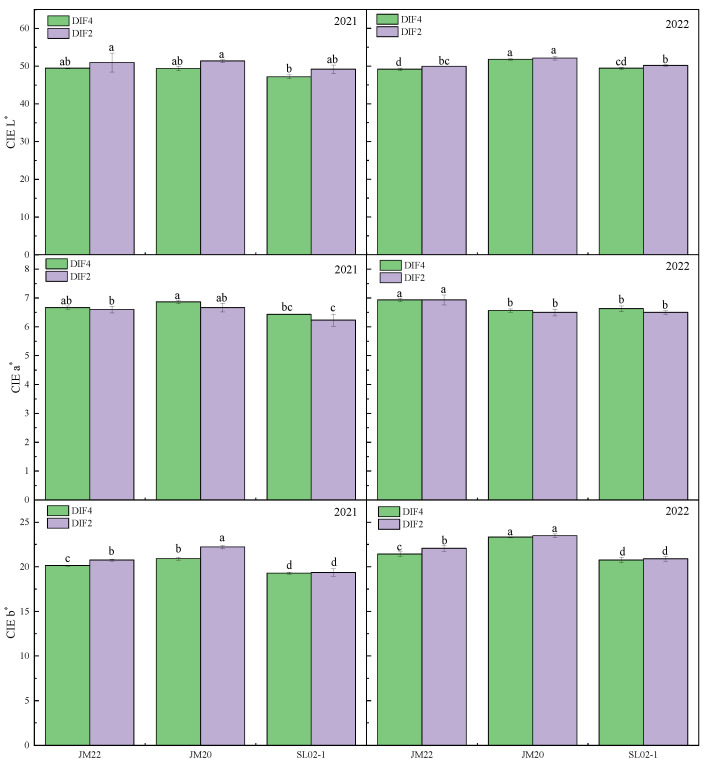
Effect of different frequencies of nitrogen application by drip irrigation on CIE L*a*b* of medium-gluten and strong-gluten wheat. JM22, JM20, and SL02-1 refer to the medium-gluten wheat variety Jimai22, and the strong-gluten wheats Jimai20 and shiluan02-1, respectively. Different lower-case letters indicate significant differences at *p* < 0.05.

**Figure 4 plants-12-03974-f004:**
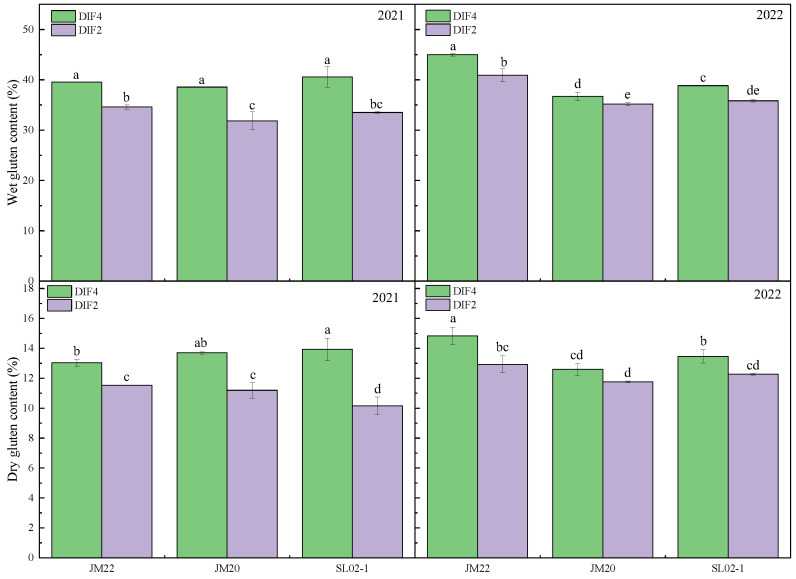
Effect of different frequencies of nitrogen application by drip irrigation on wet/dry gluten content of medium-gluten and strong-gluten wheat. JM22, JM20, and SL02-1 refer to the medium-gluten wheat variety Jimai22, and the strong-gluten wheats Jimai20 and shiluan02-1, respectively. Different lower-case letters indicate significant differences at *p* < 0.05.

**Figure 5 plants-12-03974-f005:**
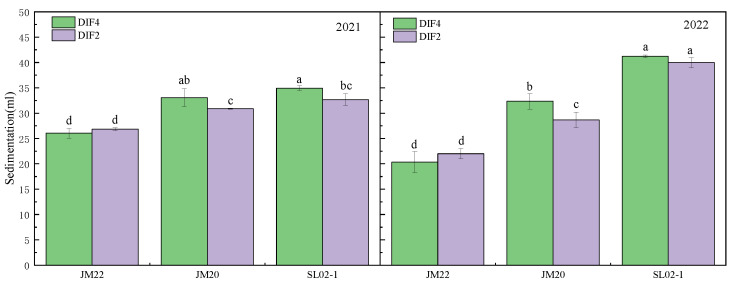
Effect of different frequencies of nitrogen application by drip irrigation on sedimentation of medium-gluten and strong-gluten wheat. JM22, JM20, and SL02-1 refer to the medium-gluten wheat variety Jimai22, and the strong-gluten wheats Jimai20 and shiluan02-1, respectively. Different lower-case letters indicate significant differences at *p* < 0.05.

**Figure 6 plants-12-03974-f006:**
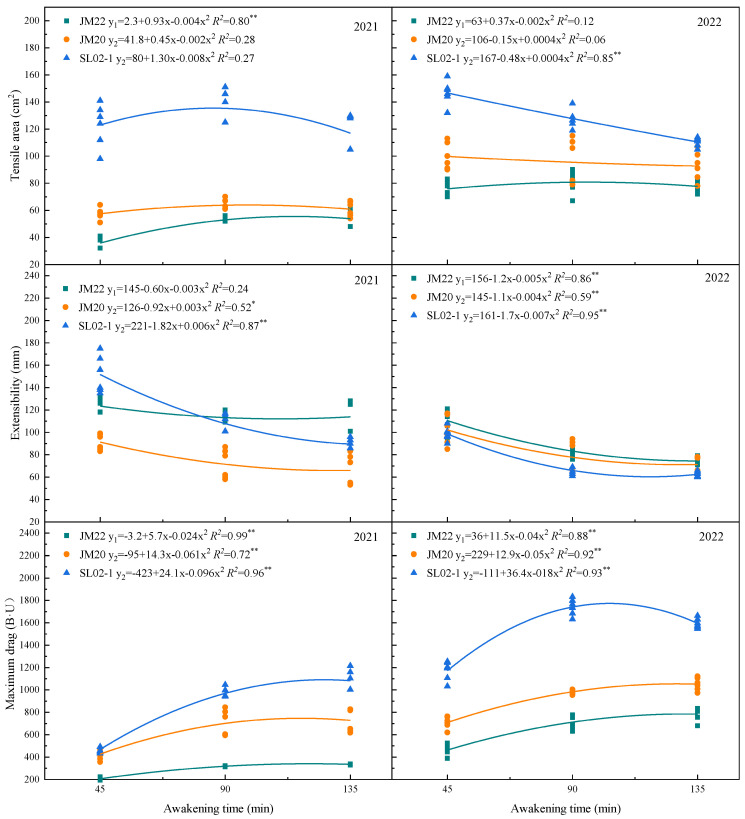
Trend of key tensile parameters over time. JM22, JM20, and SL02-1 refer to the medium-gluten wheat variety Jimai22, and the strong-gluten wheats Jimai20 and shiluan02-1, respectively. * indicates a significant difference at *p* < 0.05 and ** indicates a significant difference at *p* < 0.01.

**Figure 7 plants-12-03974-f007:**
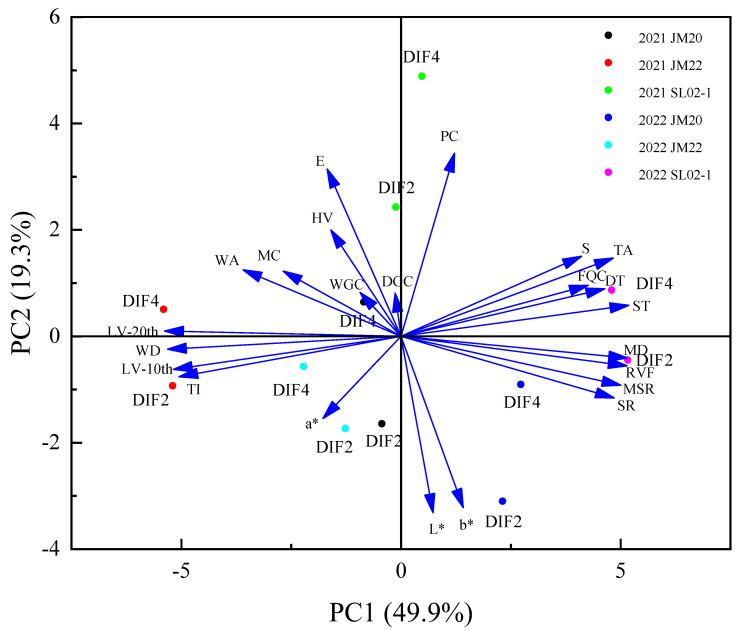
Principal component analysis of quality traits of wheat grain and its flour. HV stands for hardness value; PC stands for protein content; WGC means wet gluten content; DGC means dry gluten content; S means sedimentation; L*, a*, and b* represent CIE L*, CIE a*, and CIE b*, respectively; WA stands for water absorption; MC stands for moisture content; DT stands for development time; ST stands for stability time; TI stands for tolerance index; LV-10th stands for lowering value in the 10th minute; WD stands for weakening degree; LV-20th stands for lowering value in the 20th minute; FQC stands for flour quality index; TA stands for tensile area; RVF stands for resistance value at 5 cm; E stands for extensibility; MD stands for maximum drag; SR stands for stretch ratio; MSR stands for maximum stretch ratio.

**Figure 8 plants-12-03974-f008:**
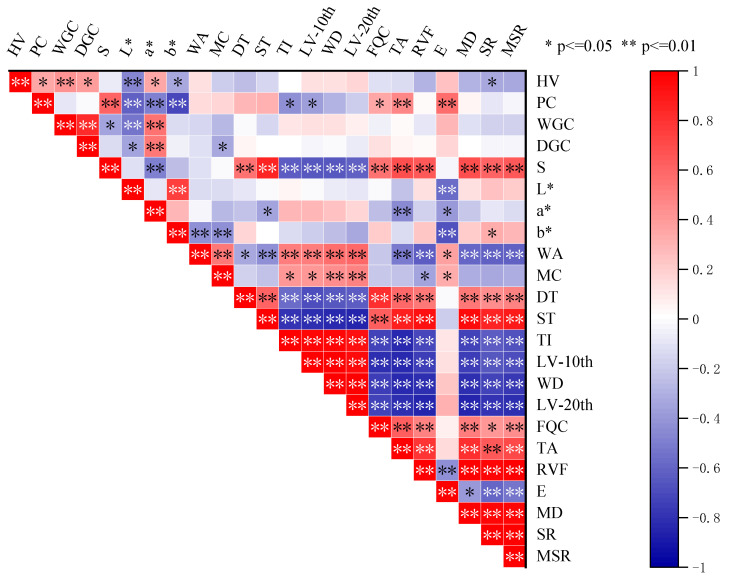
Correlation analysis of quality traits of wheat grain and its flour. HV stands for hardness value; PC stands for protein content; WGC means wet gluten content; DGC means dry gluten content; S means sedimentation; L*, a*, and b* represent CIE L*, CIE a*, and CIE b*, respectively; WA stands for water absorption; MC stands for moisture content; DT stands for development time; ST stands for stability time; TI stands for tolerance index; LV-10th stands for lowering value in the 10th minute; WD stands for weakening degree; LV-20th stands for lowering value in the 20th minute; FQC stands for flour quality index; TA stands for tensile area; RVF stands for resistance value at 5 cm; E stands for extensibility; MD stands for maximum drag; SR stands for stretch ratio; MSR stands for maximum stretch ratio. * denotes significant differences at *p* ≤ 0.05, ** denotes significant differences at *p* ≤ 0.01.

**Table 1 plants-12-03974-t001:** Interaction effects between year, species, and frequency.

Factor	Protein Content	Hardness Value	Avg CIE L*	Avg CIE a*	Avg CIE b*	Wet Gluten Content	Dry Gluten Content	Sedimentation Value
Year	**	ns	**	*	**	**	**	ns
Variety	**	ns	**	**	**	**	**	**
Frequency	**	**	**	**	**	**	**	**
Year * Variety	**	*	**	**	*	**	**	**
Year * Frequency	*	ns	*	ns	*	**	**	ns
Variety * Frequency	ns	ns	ns	ns	**	ns	*	**
Year * Variety * Frequency	*	ns	ns	ns	**	*	**	ns

* indicates a significant difference at *p* < 0.05, ** indicates a significant difference at *p* < 0.01, and ns indicates no significant difference.

**Table 2 plants-12-03974-t002:** Effect of different frequency of nitrogen application by drip irrigation on flour parameters.

Year	Variety	Drip Irrigation Fertilization Frequency	Water Absorption (%)	Moisture Content (%)	Development Time (min)	Stability Time (min)	Tolerance Index	Lowering Value in the 10th Minute (B·U)	Weakening Degree	Lowering Value in the 20th Minute (B·U)	FQC
	JM22	DIF4	54.4 b	12.8 a	1.4 b	1.7 d	245.0 a	306.0 a	331.0 a	372.7 a	19.5 d
		DIF2	60.1 a	12.7 a	1.7 b	1.9 d	262.0 a	292.0 a	316.7 a	362.0 a	22.0 d
2021	JM20	DIF4	55.8 b	10.8 b	2.4 ab	6.4 c	94.0 b	134.0 b	158.5 b	185.7 b	45.5 b
		DIF2	51.1 c	11.6 ab	2.1 b	5.8 c	103.0 b	136.0 b	162.5 b	203.0 b	37.0 c
	SL02-1	DIF4	54.0 bc	11.6 ab	3.9 a	9.2 a	70.3 c	79.7 c	139.0 b	180.7 b	64.7 a
		DIF2	53.5 bc	11.6 ab	2.5 ab	8.0 b	60.0 c	119.0 b	144.0 b	200.0 b	36.0 c
	JM22	DIF4	49.4 ab	9.2 b	1.9 b	3.5 d	142.3 a	191.3 a	211.7 a	232.7 a	28.7 c
		DIF2	52.1 a	10.1 ab	2.1 b	3.7 d	118.7 a	158.7 a	170.0 b	183.7 b	34.0 bc
2022	JM20	DIF4	52.3 a	10.0 ab	4.0 a	11.5 b	48.7 c	63.7 bc	91.0 cd	101.3 cd	72.0 a
		DIF2	47.1 b	9.0 b	2.9 ab	8.7 c	84.3 b	99.7 b	116.3 c	129.3 c	49.0 b
	SL02-1	DIF4	49.9 ab	9.7 ab	3.1 ab	18.9 a	37.3 c	53.3 c	59.0 d	60.3 d	55.7 ab
		DIF2	49.8 ab	11.0 a	3.0 ab	18.8 a	33.3 c	62.0 bc	77.3 d	78.0 d	49.3 b
Year			**	**	*	**	**	**	**	**	**
Variety			**	*	**	**	**	**	**	**	**
Frequency			ns	ns	ns	**	ns	ns	ns	ns	**
Year * Variety			ns	ns	ns	**	**	**	**	**	*
Year * Frequency			ns	ns	ns	ns	ns	ns	ns	ns	ns
Variety * Frequency			**	ns	ns	**	**	**	**	**	**
Year * Variety * Frequency			ns	*	ns	**	*	ns	ns	ns	*

JM22, JM20, and SL02-1 refer to the medium-gluten wheat variety Jimai22, and the strong-gluten wheats Jimai20 and Shiluan02-1, respectively. Different lower-case letters indicate significant differences at *p* < 0.05. * indicates a significant difference at *p* < 0.05, ** indicates a significant difference at *p* < 0.01, and ns indicates no significant difference.

**Table 3 plants-12-03974-t003:** Effect of drip irrigation nitrogen application frequency on gluten index of Jimai22.

Year	Treatment	Gluten Index (%)
2021	DIF4	80.5 a
	DIF2	86.2 a
2022	DIF4	57.7 b
	DIF2	71.0 a

Different lower-case letters indicate significant differences at *p* < 0.05.

**Table 4 plants-12-03974-t004:** Effect of different number of drip irrigation nitrogen applications on stretching parameters of medium-gluten and strong-gluten wheat.

Year	Variety	Drip Irrigation Fertilization Frequency	Tensile Area (cm^2^)	Resistance Value at 5 cm (B·U)	Extensibility (mm)	Maximum Drag (B·U)	Stretch Ratio	Maximum Stretch Ratio
2021	JM22	DIF4	39.7 bc	172.7 d	129.0 c	200.7 d	1.6 e	1.6 d
		DIF2	32.0 c	178.0 d	118.0 d	210.7 d	1.7 e	1.8 d
	JM20	DIF4	58.0 b	380.7 c	97.7 e	385.0 c	4.0 b	4.0 b
		DIF2	57.0 b	468.7 a	85.0 f	468.7 ab	5.6 a	5.6 a
	SL02-1	DIF4	129.0 a	417.3 b	165.7 a	445.7 b	2.6 d	3.4 c
		DIF2	117.0 a	419.7 b	137.7 b	491.7 a	3.2 c	3.6 c
2022	JM22	DIF4	78.0 cd	437.0 d	118.3 a	441.0 d	3.7 d	3.8 e
		DIF2	73.7 d	480.7 d	102.7 bc	485.0 d	4.7 d	4.8 de
	JM20	DIF4	107.7 b	627.0 c	113.0 ab	695.7 c	5.6 d	6.3 d
		DIF2	92.0 c	716.3 c	91.3 c	723.0 c	8.0 c	8.1 c
	SL02-1	DIF4	147.7 a	1044.3 b	101.7 bc	1111.7 b	10.4 b	11.0 b
		DIF2	145.7 a	1171.3 a	95.7 c	1233.3 a	12.3 a	13.0 a
Year			**	**	**	**	**	**
Variety			**	**	**	**	**	**
Frequency			*	**	**	**	**	**
Year * Variety			*	**	**	**	**	**
Year * Frequency			ns	*	ns	ns	*	*
Variety * Frequency			ns	ns	ns	ns	*	ns
Year * Variety * Frequency			ns	ns	**	ns	ns	ns

JM22, JM20, and SL02-1 refer to the medium-gluten wheat variety Jimai22, and the strong-gluten wheats Jimai20 and shiluan02-1, respectively. Different lower-case letters indicate significant differences at *p* < 0.05. * indicates a significant difference at *p* < 0.05, ** indicates a significant difference at *p* < 0.01, and ns indicates no significant difference.

**Table 5 plants-12-03974-t005:** Irrigation and nitrogen application during the critical fertility period of wheat.

Amount of Water and Nitrogen Fertilizer	Treatment	Wheat Growth Period					
		Seeding	Jointing	Booting	Flowering	Filling	Total
N application (kg·ha^−1^)	DIF4	90	30	30	30	30	210
	DIF2	90	60		60		210
Irrigation volumes (mm)	DIF4		30	30	30	30	120
	DIF2		60		60		120

## Data Availability

Data will be made available on request.

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
