# Peer review of "Enhancing Wheat Gluten Content and Processing Quality: An Analysis of Drip Irrigation Nitrogen Frequency"

_plants, 2023, doi:10.3390/plants12233974_

Round 1

Reviewer 1 Report

Comments and Suggestions for Authors

In this paper, the different drip irrigation nitrogen frequency was applied to investigate the effects of this technique on the dough quality traits of medium-gluten and strong-gluten wheats; and the relationship between the dough quality traits. However, the manuscript was poorly organized, extensive revision is required before it’s ready for publication. In the Introduction part, only the background of wheat grain quality is presented. Neither the water irrigation nor the nitrogen application was introduced there. From the title of this paper “Enhancing Wheat Gluten Content and Processing Quality: an Analysis of Drip Irrigation Nitrogen Frequency”, shouldn’t be the latest research on the water and nitrogen management included in this part? The last paragraph of the introduction failed to give us the urgency and innovation of this paper, Besides, the most important technique (drip irrigation nitrogen frequency) even wasn’t described in the M&M! Only one reference was cited there. The discussion part was only the simple comparison between the current results and the previous study. It’s not the scientific writing. Please re-write it! In the end,  The major revision is essential before resubmission. 

Comments on the Quality of English Language

The extensive language editing is required for this draft.

Author Response

Response to Review 1 Comments

Dear reviewer:

Thank you for your busy schedule to provide valuable comments on this manuscript, which have led to further revision and improvement of the manuscript. In response to your comments, we have revised and improved the introduction, materials and methods, and discussion sections. We have marked the changes in red in the original text. The main corrections in the paper and the responds to the reviewer's comments are as flowing: The response has been marked in red.

Comments:

In this paper, the different drip irrigation nitrogen frequency was applied to investigate the effects of this technique on the dough quality traits of medium-gluten and strong-gluten wheats; and the relationship between the dough quality traits. However, the manuscript was poorly organized, extensive revision is required before it’s ready for publication. In the Introduction part, only the background of wheat grain quality is presented. Neither the water irrigation nor the nitrogen application was introduced there. From the title of this paper “Enhancing Wheat Gluten Content and Processing Quality: an Analysis of Drip Irrigation Nitrogen Frequency”, shouldn’t be the latest research on the water and nitrogen management included in this part? The last paragraph of the introduction failed to give us the urgency and innovation of this paper, Besides, the most important technique (drip irrigation nitrogen frequency) even wasn’t described in the M&M! Only one reference was cited there. The discussion part was only the simple comparison between the current results and the previous study. It’s not the scientific writing. Please re-write it! In the end, the major revision is essential before resubmission.

Response:

In the introduction section, we have added an introduction to traditional irrigation and nitrogen application, as well as the advantages of drip irrigation, and in the last paragraph we have also explained the innovation and necessity of split nitrogen application with drip irrigation. In the Materials and Methods section. I rewrote the information about the frequency of nitrogen application by drip irrigation because it was not written clearly. Discussion: There are some parts that compare the previous studies, which is not scientific, I rewrote these parts

Reviewer 2 Report

Comments and Suggestions for Authors

The following are the revisions needful:

1. L13-14: Are you sure that drip irrigation is a recent technology and that nitrogen application under such a system has not been optimized? Drip irrigation has been commercially used for the last two decades. 

2. L 39: Add the following recent reference: https://doi.org/10.1016/j.heliyon.2023.e13997

3. In the introduction part, the need for the research is missing. The author needs to add one paragraph in the introduction section on this.

4. The results part is well-written with the suitable depiction of figures and tables.

5.  How did the DIF4 significantly improve almost all the quality parameters over DIF2 when the amount of N was the same? Your results are looking like increasing the drip irrigation frequency continuously increasing the quality attributes. Is it so? Moreover, why does the author take only two levels of drip irrigation? why not DIF 3 or DIF 5??

Justify this part.

6. L 325; 340; 342: Describe in detail with suitable reference. 

7. L 363: Poorly described. Need more detailed description of the statistical design of the study, types of ANOVA, etc. 

Author Response

Response to Reviewer 2 Comments

Dear reviewer:

Thank you for your comments, which are all valuable and very helpful for revising and improving our paper. We have studied comments carefully and have made correction which we hope meet with approval. We have marked the changes in red in the original text. The main corrections in the paper and the responds to the reviewer's comments are as flowing: The response has been marked in red.

  1. L13-14: Are you sure that drip irrigation is a recent technology and that nitrogen application under such a system has not been optimized? Drip irrigation has been commercially used for the last two decades.
  2. L 39: Add the following recent reference: https://doi.org/10.1016/j.heliyon.2023.e13997
  3. In the introduction part, the need for the research is missing. The author needs to add one paragraph in the introduction section on this.
  4. The results part is well-written with the suitable depiction of figures and tables.
  5.  How did the DIF4 significantly improve almost all the quality parameters over DIF2 when the amount of N was the same? Your results are looking like increasing the drip irrigation frequency continuously increasing the quality attributes. Is it so? Moreover, why does the author take only two levels of drip irrigation? why not DIF 3 or DIF 5?? Justify this part.
  6. L 325; 340; 342: Describe in detail with suitable reference.
  7. L 363: Poorly described. Need more detailed description of the statistical design of the study, types of ANOVA, etc.

Response:

  1. I'm very sorry I wrote it wrong, it wasn't supposed to be the result I was trying to convey. Changes have been made in the manuscript accordingly. A review describes water and fertilizer integration in this way: in the 1960s, water and fertilizer integration technology originated in Israel, and as of 2018, 90% of Israel's agricultural production has realized water and fertilizer integration and intelligent management, in the 1970s, China introduced drip irrigation equipment from Mexico, and at present, China's Xinjiang's cotton under-membrane drip irrigation technology has reached the international leading level, and in recent years, the country has vigorously promoted the application of This technology, the popularization area continues to increase[1].
  2. I have cited the literature at L39. The use of POLY4 in India can increase economic efficiency, yield and quality, a very interesting study, I have searched from Chinese databases and found that there is no study on its use as a crop fertilizer, very innovative.
  3. Improved introduction.
  4. I have also made appropriate changes to the results section.
  5. The enhancement of protein content and wet gluten content with the same nitrogen was significant and was evident in the manuscript, as almost all showed variability. DIF4 irrigated and applied nitrogen during two more periods than DIF2 (pregnancy and irrigated, although the amount of nitrogen and irrigation per application was reduced). The grouting period is the period when wheat sets seed and the kernels become plump and finally mature. In our previous study, we showed that increasing the frequency of nitrogen application by drip irrigation resulted in improved irrigating capacity, prolonged green-up time during the irrigating period, and increased grain weight [2]. A recent study suggests that moderately delaying leaf senescence may inhibit protein degradation [3]. In this study, the differences in flour quality and tensile quality between DIF4 (water and nitrogen applied at the nodulation, ear, flowering and irrigation stages) and DIF2 (water and nitrogen applied at the nodulation and flowering stages) were not significant, and nitrogen applied by high-frequency drip irrigation improved the flour quality and tensile quality of the strong-gluten varieties but not the medium-gluten variety, Jimai 22, which was not significantly reduced (p>0.05). In this study, DIF3 (water and nitrogen applied at nodulation, flowering, and grouting stages) was set up, and the medium-gluten variety Jimai 22 had the highest yield of DIF4 and the lowest of DIF2, while the strong-gluten varieties Jimai 20 and Shi Luan 02-1 had the highest yield of DIF2 and the lowest of DIF4, and both were non-significant [2]. Therefore, only DIF4 and DIF2 were considered in this manuscript. DIF5 was not set in this study. The period of nodulation, flowering, gestation, and grouting are the more important periods of wheat. Farmers usually choose to apply all the Nitrogen fertilizers at one time during nodulation, or to apply Nitrogen at the period of nodulation and flowering. However, there are cases of applying fertilizers at other periods. DIF5 was not set in this study.
  6. L 325: Many articles on Kjeldahl nitrogen determination have been written in passing, with no citations [4,5]. I have enriched the presentation a little in the text. I also improved L340 and L342.
  7. I compared what and how some of the literature is written up and I rewrote this section to make it more complete.

Reference:

  1. Shi, X.; Pei, X.; Dang, J. and Zhang, D. Research progress on high-yield, high-quality, high-efficiency and ecology cultivation of wheat micro-sprinkling and drip fertigation. Crops. 2022, (1): 1-10. doi:10.16035/j.issn.1001-7283.2022.01.001
  2. Hao, T.; Zhu, Z.; Zhang, Y.; Liu, S.; Xu, Y.; Xu, X. and Zhao, C. Effects of drip irrigation and fertilization frequency on yield, water and nitrogen use efficiency of medium and strong gluten wheat in the Huang-Huai-Hai plain of China. Agronomy. 2023, 13(6). doi:10.3390/agronomy13061564
  3. Zhou, M. and Yang, J. Delaying or promoting? Manipulation of leaf senescence to improve crop yield and quality. Planta. 2023, 258(3): 48. doi:10.1007/s00425-023-04204-1
  4. Martre, P.; Porter, J.R.; Jamieson, P.D. and Triboi, E. Modeling grain nitrogen accumulation and protein composition to understand the sink/source regulations of nitrogen remobilization for wheat. Plant Physiol. 2003, 133(4): 1959-1967. doi:10.1104/pp.103.030585
  5. Rathke, G.W.; Christen, O. and Diepenbrock, W. Effects of nitrogen source and rate on productivity and quality of winter oilseed rape (Brassica napus L.) grown in different crop rotations. Field Crops Res.2005, 94(2-3): 103-113. doi:10.1016/j.fcr.2004.11.010

Reviewer 3 Report

Comments and Suggestions for Authors

Abstract

Line 15-20: should rewrite and separate the sentence. It is difficult to understand. Should have a recommendation for the following research.

Line 19: check the information on nitrogen application (210 kg ha-1) and total irrigation (120 mm); it is incorrect because of the different information in Table 5.

Introduction

What is traditional irrigation applied for wheat in China? Using the new technique will have perfect results, but in the introduction, you have not mentioned the urgency and effectiveness of applying the drip irrigation technique you used compared to the traditional method. It is possible that applying this technique will improve wheat quality, but only to a small extent. In an enormous scope, how effective will the investment and application of this measure be?

The introduction still lacks information compared to the topic given and needs to be made more explicit.

Materials and methods

Lines 303-304 should be rewritten because it confuses readers: “The seeds were sown with the fertilizers.”

305: The primary wheat seedlings were about 2.2 million plant ha-1. Should be given the number of seeds better.

Table 5. Should add more detail/information about the agronomy practices such as plow, organic fertilizer (if have), fertilizers application rate, and time of P2O5 and K2O. Sowing and harvest date…

What is the unit you used for the amount of water and nitrogen fertilizer (Table 5)?

You say to see the previous article, but what is different between this manuscript and your last paper (as you mentioned) for the N application and irrigation volume? The previous article shows that N application at seeding is 90 kg ha-1 and 0 mm irrigation volume, but this manuscript is the opposite. I don’t understand, please explain. The “total” column is also the same.

Why did you apply K2O one time at the sowing time? Could you explain some information about it, please?

Line 337: Correct “WGC” instead of “GGC”.

Protein content and Sedimentation value tests: should be a more detailed description of the method.

Data analysis: should detail more influence between the factors.

Result

Add the supplementation table for the parameters in the table 1.

Line 71: What does seed color mean? “A highly significant effect on seed color”?

According to Table 1: P<0.01, it is a highly significant effect; why are you writing “could have”? I don’t understand this one. Should rewrite the sentence.

Line 80: should you add P<0.05 and P<0.01 because “at the 0.05” and “at the 0.01” are different?

Line 86-88. This sentence isn’t clear: The protein content of the second year was lower than the first year, which might have been antagonized by the yield, which was higher than the first year… please indicate the results of grain yield on the paper; it will be easy understand for readers. And also detailed in the discussion task.

Line 89: What is the effect/relation of “the sowing date of the second year was delayed by successive rainy days” on lower protein content in the second year compared to the first year?

Lines 100, 105, 110, 128 and 133, 159, 160….: add P<0.05.

Discussion

Line 293: correct citation format “(Peng et al. 2022)”, [number].

Conclusion

Lines 368-369: the conclusion is unclear; this sentence means conclude for only Jimai22; please detail another variety and different drip irrigation applied.

Should have the comments/recommendations from the results achieved.

Reference

They should be added the DOI in.

Author Response

Response to Reviewer 3 Comments

Dear reviewer:

Thank you for your comments, which are all valuable and very helpful for revising and improving our paper. We have studied comments carefully and have made correction which we hope meet with approval. We have marked the changes in red in the original text. The main corrections in the paper and the responds to the reviewer's comments are as flowing:

The response has been marked in red.

Abstract

  • Line 15-20: should rewrite and separate the sentence. It is difficult to understand. Should have a recommendation for the following research.
  • Line 19: check the information on nitrogen application (210 kg ha-1) and total irrigation (120 mm); it is incorrect because of the different information in Table 5.

Response:

  • I split L15-20 into two sentences to make it more accessible.
  • Because of my carelessness, the information in Table V was put in backwards and has been corrected.

Introduction

What is traditional irrigation applied for wheat in China? Using the new technique will have perfect results, but in the introduction, you have not mentioned the urgency and effectiveness of applying the drip irrigation technique you used compared to the traditional method. It is possible that applying this technique will improve wheat quality, but only to a small extent. In an enormous scope, how effective will the investment and application of this measure be?

The introduction still lacks information compared to the topic given and needs to be made more explicit.

Response:

I have rewritten the introduction. It describes the traditional methods of fertilization and irrigation, explains the advantages of drip irrigation, and further writes in the last paragraph about the importance of conducting this research.

Materials and methods

  • Lines 303-304 should be rewritten because it confuses readers: “The seeds were sown with the fertilizers.”

(2) 305: The primary wheat seedlings were about 2.2 million plant ha-1. Should be given the number of seeds better.

(3) Table 5. Should add more detail/information about the agronomy practices such as plow, organic fertilizer (if have), fertilizers application rate, and time of P2O5 and K2O. Sowing and harvest date…

(4) What is the unit you used for the amount of water and nitrogen fertilizer (Table 5)?

(5) You say to see the previous article, but what is different between this manuscript and your last paper (as you mentioned) for the N application and irrigation volume? The previous article shows that N application at seeding is 90 kg ha-1 and 0 mm irrigation volume, but this manuscript is the opposite. I don’t understand, please explain. The “total” column is also the same.

(6) Why did you apply K2O one time at the sowing time? Could you explain some information about it, please?

(7) Line 337: Correct “WGC” instead of “GGC”.

(8) Protein content and Sedimentation value tests: should be a more detailed description of the method.

(9) Data analysis: should detail more influence between the factors.

Response:

  • I rewrote the sentence to make it as understandable as possible for the reader.
  • I am very sorry that we are not able to provide information on the number of seeds, we use a small precision seed and fertilizer co-seeder, due to the different varieties and the different thousand seed weights, the number and weight of seeds sown is calculated by the thousand seed weights and the seedling emergence rate to make sure that the number of seeds is the same, but I don't keep a record of this information, I just use it as an aid to the sowing of the seeds.
  • The time of P2O5 and K2O, sowing and harvest date have been described in detail in 4.1.
  • Nitrogen fertilizer in kg·ha-1and water in mm were added to the table.
  • I am very sorry and remorseful that I made a mistake in Table 5, where I reversed the nitrogen application and irrigation rates, which has been corrected in the text.
  • Both P2O5and K2O were applied in a single pre-planting application [1,2]. In the countryside, P and K fertilizers are applied at once before sowing, perhaps through drip irrigation technology to achieve potash fertilizer follow-up is also a good direction.
  • It has been corrected.
  • Protein content and Sedimentation value tests have been described in more detail.
  • Based on the previous literature, we provide a more detailed description of the ''Data analysis'' section.

Result

  • Add the supplementation table for the parameters in the table 1.
  • Line 71: What does seed color mean? “A highly significant effect on seed color”?

According to Table 1: P<0.01, it is a highly significant effect; why are you writing “could have”? I don’t understand this one. Should rewrite the sentence.

  • Line 80: should you add P<0.05 and P<0.01 because “at the 0.05” and “at the 0.01” are different?
  • Line 86-88. This sentence isn’t clear: The protein content of the second year was lower than the first year, which might have been antagonized by the yield, which was higher than the first year… please indicate the results of grain yield on the paper; it will be easy understand for readers. And also detailed in the discussion task.
  • Line 89: What is the effect/relation of “the sowing date of the second year was delayed by successive rainy days” on lower protein content in the second year compared to the first year?
  • Lines 100, 105, 110, 128 and 133, 159, 160….: add P<0.05.

 Response:

  • The parameters in Table 1 don't seem to require much information, which I'm a little confused about. If it's an ANOVA, it's already expressed in the figure below, and the F-value in the reciprocal analysis is not required. No changes have been made here.
  • Seed color is expressed in CIEL*a*b*. "could have" was a mistake in my writing, which conveyed the wrong meaning and has been corrected. I have rewritten the sentence.
  •  
  • I added the magnitude of change because it is more concise. I have noted in the results section "probably antagonized by yield" and cited the literature, and will not discuss this phenomenon in the discussion section.
  • That sentence seemed useless here, so I moved it to Materials and Methods.
  • It has been modified.

Discussion

Line 293: correct citation format “(Peng et al. 2022)”, [number].

Response:

Corrected.

Conclusion

  • Lines 368-369: the conclusion is unclear; this sentence means conclude for only Jimai22; please detail another variety and different drip irrigation applied.
  • Should have the comments/recommendations from the results achieved.

Response:

(1) I've broken up the sentences to make them clearer. The first sentence summarizes the three varieties, and the second sentence only summarizes the gluten index of the medium-gluten variety Jimai22, since the strong-gluten variety has a gluten index of nearly 100%.

(2) Based on the experimental results, appropriate recommendations were made for the purpose of achieving high protein content or high processing quality.

Reference

They should be added the DOI in.

Response:

Corrected.

Reference

  1. Li, J.; Wang, Z.; Song, Y.; Li, J. and Zhang, Y. Effects of reducing nitrogen application rate under different irrigation methods on grain yield, water and nitrogen utilization in winter wheat. Agronomy. 2022, 12(8): 1-16. doi:10.3390/agronomy12081835
  2. Huang, P.; Zhang, J.; Zhu, A.; Xin, X.; Zhang, C.; Ma, D.; Yang, S.; Mirza, Z. and Wu, S. Coupled water and nitrogen (N) management as a key strategy for the mitigation of gaseous n losses in the Huang-Huai-Hai plain. Biol. Fertil. Soils. 2014, 51(3): 333-342. doi:10.1007/s00374-014-0981-0

Round 2

Reviewer 1 Report

Comments and Suggestions for Authors

N/A

Comments on the Quality of English Language

N/A

Author Response

Thank you for your previous valuable comments, and I am very sorry that the last modification did not meet your satisfaction. In this revision, there is no revision comment shown in the system. If you have any other specific comments on this manuscript, it will definitely be greatly improved.

Reviewer 2 Report

Comments and Suggestions for Authors

The authors improved the paper as per the suggestions given. It may be accepted now.

Author Response

Dear reviewer

Thank you very much for your previous comments, which were a great enhancement to our manuscript, and your approval of this manuscript is greatly appreciated. The current revision system does not show that you have a revision comment. If you still have questions about this manuscript, please feel free to comment. Thank you again.